# Experimental Models to Study Immune Dysfunction in the Pathogenesis of Parkinson’s Disease

**DOI:** 10.3390/ijms25084330

**Published:** 2024-04-14

**Authors:** Jasna Saponjic, Rebeca Mejías, Neda Nikolovski, Milorad Dragic, Asuman Canak, Stamatia Papoutsopoulou, Yasemin Gürsoy-Özdemir, Kari E. Fladmark, Panagiotis Ntavaroukas, Nuray Bayar Muluk, Milica Zeljkovic Jovanovic, Ángela Fontán-Lozano, Cristoforo Comi, Franca Marino

**Affiliations:** 1Department of Neurobiology, Institute of Biological Research “Sinisa Stankovic”, National Institute of the Republic of Serbia, University of Belgrade, 11108 Belgrade, Serbia; 2Department of Physiology, School of Biology, University of Seville, 41012 Seville, Spain; rmejias@us.es (R.M.); afontan@us.es (Á.F.-L.); 3Instituto de Biomedicina de Sevilla, IBiS, Hospital Universitario Virgen del Rocío, CSIC, Universidad de Sevilla, 41013 Seville, Spain; 4Department of Immunology, Institute for Biological Research “Siniša Stanković”, National Institute of the Republic of Serbia, University of Belgrade, 11108 Belgrade, Serbia; neda.djedovic@ibiss.bg.ac.rs; 5Laboratory for Neurobiology, Department for General Physiology and Biophysics, Faculty of Biology, University of Belgrade, 11000 Belgrade, Serbia; milorad.dragic@bio.bg.ac.rs (M.D.); milica.zeljkovic@bio.bg.ac.rs (M.Z.J.); 6Department of Molecular Biology and Endocrinology, Vinca Institute of Nuclear Sciences–National Institute of the Republic of Serbia, University of Belgrade, 11351 Belgrade, Serbia; 7Department of Medical Services and Techniques, Vocational School of Health Services, Recep Tayyip Erdogan University, Rize 53100, Turkey; asuman.canak@gmail.com; 8Department of Biochemistry and Biotechnology, Faculty of Health Sciences, University of Thessaly, Biopolis, 41500 Larisa, Greece; stapapou@uth.gr (S.P.); pntavaroukas@uth.gr (P.N.); 9Department of Neurology, School of Medicine, Koç University, Istanbul 34010, Turkey; ygursoy@ku.edu.tr; 10Department of Biological Science, University of Bergen, 5020 Bergen, Norway; kari.fladmark@uib.no; 11Department of Otorhinolaryngology, Faculty of Medicine, Kirikkale University, Kirikkale 71450, Turkey; nbayarmuluk@kku.edu.tr; 12Neurology Unit, Department of Translational Medicine, S. Andrea Hospital, University of Piemonte Orientale, 13100 Vercelli, Italy; cristoforo.comi@med.uniupo.it; 13Center for Research in Medical Pharmacology, School of Medicine, University of Insubria, 21100 Varese, Italy; franca.marino@uninsubria.it

**Keywords:** Parkinson’s disease, immune systems, neuroinflammation, neurodegeneration, peripheral immune cells, animal models, cellular models

## Abstract

Parkinson’s disease (PD) is a chronic, age-related, progressive multisystem disease associated with neuroinflammation and immune dysfunction. This review discusses the methodological approaches used to study the changes in central and peripheral immunity in PD, the advantages and limitations of the techniques, and their applicability to humans. Although a single animal model cannot replicate all pathological features of the human disease, neuroinflammation is present in most animal models of PD and plays a critical role in understanding the involvement of the immune system (IS) in the pathogenesis of PD. The IS and its interactions with different cell types in the central nervous system (CNS) play an important role in the pathogenesis of PD. Even though culture models do not fully reflect the complexity of disease progression, they are limited in their ability to mimic long-term effects and need validation through in vivo studies. They are an indispensable tool for understanding the interplay between the IS and the pathogenesis of this disease. Understanding the immune-mediated mechanisms may lead to potential therapeutic targets for the treatment of PD. We believe that the development of methodological guidelines for experiments with animal models and PD patients is crucial to ensure the validity and consistency of the results.

## 1. Introduction

Parkinson’s disease (PD) is the second most common and fastest growing neurodegenerative disease of the elderly, with a prevalence of 1–3% of people over 65 years of age, already affecting more than 6.2 million people worldwide and expected to increase to more than 12 million by 2040 [1,2,3].

PD is a chronic, heterogeneous, and progressive neurodegenerative disease pathologically characterized by intracellular aggregates of α-synuclein (α-syn) in Lewy bodies (LBs) and neurites and loss of dopaminergic (DA) neurons in the substantia nigra (SN), which is responsible for the onset of motor symptoms [4,5].

Moreover, PD is a multisystem disease in which α-syn pathology and neuronal degeneration also occur in non-dopaminergic pathways in the central nervous system (CNS) and peripheral nervous system (PNS), which may precede degeneration in SN and is associated with a variety of non-motor symptoms, such as hyposmia, constipation, fatigue, depression, and sleep disorders [4]. Motor impairment, as one of the main pathological features of PD indicating the onset of clinical PD and encompassing a spectrum of movement and postural abnormalities (bradykinesia, rigidity, resting tremor, and postural and gait problems), is primarily associated with a progressive loss of DA neurons of the SN pars compacta (SNpc) and the consequent reduction in striatal DA levels [6]. This occurs relatively late, when DA neurodegeneration is around 60–80%, limiting the possibility of effective medical treatment [6]. However, PD pathology goes beyond DA pathway loss and also involves the serotonergic, noradrenergic, cholinergic, GABAergic, and glutamatergic systems, which are associated with the non-motor symptoms of PD that can precede motor symptoms by years or even decades, which provides a unique opportunity to investigate the progression of PD, identify potential prodromal markers, perform presymptomatic investigations, and potentially intervene therapeutically at an early stage [7].

At present, the origin of the initial α-syn aggregates, which appear to propagate from cell to cell in a prion-like manner, remains unclear. In this sense, the existence of two PD subtypes has been recently hypothesized: “brain-first” and “body-first” PD, in which neurodegeneration begins in either the CNS or PNS [8]. The theoretical connectome model (SOC) for Lewy body disease is based on the hypothesis that, in the majority of patients, the initial α-syn pathology originates at a single site and spreads from there. The most common sites of origin are hypothesized to be the olfactory system and the enteric nervous system. The SOC model predicts that, unlike olfactory-first pathology, which leads to the “brain first” subtype of PD with fewer non-motor symptoms before diagnosis, “gut first” pathology leads to a “body first” clinical subtype of PD. This latter subtype is characterized by older patients, symmetric degeneration of DA neurons, an increased risk of dementia, autonomic prodromal symptoms, and REM sleep behavior disorders (RBDs) [9].

Although it has been hypothesized that there are two PD subtypes, “brain-first” and “body-first,” there are currently no definitive diagnostic criteria for the prodromal stage of PD, apart from non-motor risk markers, such as polysomnographically proven isolated RBD (iRBD), hyposmia, constipation, orthostatic hypotension, erectile dysfunction, urinary dysfunction, and depression. iRBD, a parasomnia characterized by enactment out of dreams due to lack of muscle atonia, is currently considered the most important prodromal marker for PD and other synucleinopathies [10].

Currently, only symptomatic therapies are available for PD, so both the establishment of early diagnostic methods and the development of disease-modifying therapies to prevent the onset or slow the progression of the disease are urgently needed. Although the non-motor symptoms of PD have attracted much attention as a clue for identifying patients in the prodromal stage of PD, the new animal models for the prodromal stage of the disease, in which dopaminergic cell loss is minimal, are essential for the development of disease-modifying therapies.

In addition, due to the difficulty of reproducing all aspects of the preclinical, prodromal, and advanced stages of PD in a single animal model, the optimal models should be selected according to the specific purpose of the experiment [11].

In the last 20 years, our knowledge of the role of neuroinflammation and peripheral immune alterations in the pathophysiology of PD has expanded rapidly. PD is now understood to be a multisystem disorder associated with neuroinflammation and immune dysfunction, along with the development of a variety of non-motor symptoms that may precede the diagnosis of the disease by decades [5].

As PD is an age-related disease, immunosenescence (age-related immune deficiency and inflammation) is an important factor in its pathogenesis. Both the innate and the adaptive immune system (IS) lose their competence with age and are also altered in PD. Inflammaging is characterized by excessive production of circulating inflammatory mediators or cytokines at low levels, such as C-reactive protein (CRP), interleukin 6 (IL6), and tumor necrosis factor (TNF), by chronically stimulated innate and adaptive immune cells.

Numerous studies using postmortem, in vitro, and animal model approaches have shown that neuroinflammation is an important pathway in the pathogenesis of PD and that this process involves both innate and adaptive immunity mechanisms [12,13,14].

Hence, understanding the immune-mediated mechanisms may lead to potential therapeutic targets for the treatment of PD.

In this review, we will discuss the methodological approaches used to study changes in central and peripheral immunity in PD using animal models. We will also discuss the advantages and limitations of the techniques and their transferability to humans. In addition, we will discuss the possible reasons for the variability in the results of different studies and possible solutions to overcome these limitations.

## 2. Animal Models to Study the Immune System in PD

The most commonly used animal models in the study of the immune system in PD are briefly described below.

### 2.1. Toxin-Induced Models of PD

Toxic models of PD aim to reproduce some aspects of the pathological and behavioral changes of the human disease in rodents and primates. They are based on the systemic or local (intracerebral) administration of specific neurotoxins that can induce selective degeneration of nigrostriatal DA neurons. The two most commonly used toxic models of PD are the classical 6-hydroxydopamine-induced (6-OHDA) model in rats [15,16,17] and the MPTP model in mice and monkeys (Figure 1) [18].

Although in both toxic models, the degeneration of nigrostriatal DA neurons is rapidly induced (2–3 days or 2–3 weeks, depending on the protocol), their common weakness is that they lack LB/α-syn pathology. In the past, both models have been used to advance successful symptomatic treatments and refine therapies for the motor symptoms of PD [19].

#### 2.1.1. 6-Hydroxydopamine-Induced (6-OHDA) Model of PD

6-OHDA is a potent and selective catecholaminergic neurotoxin identified more than 60 years ago [20] and is one of the most commonly used toxins to produce lesions of nigrostriatal DA neurons in rats and mice [21]. 6-OHDA is a structural analog of dopamine (DA) that is readily oxidized and taken up by dopamine, noradrenaline, and serotonin transporters [22]. 6-OHDA exerts its cytotoxic effects through several well-described pathways, i.e., production of reactive oxygen (RO) and nitrogen species [23] and direct inhibition of mitochondrial respiratory chain complex I and IV [24]. Since 6-OHDA does not cross the blood–brain barrier (BBB), the injections are administered directly into the brain, i.e., intracerebrally, to the desired anatomical region. Specifically, 6-OHDA can be administered directly into the substantia nigra pars compacta (SNpc), medial forebrain bundle (MBF), striatum (CPu) [18], or rarely intraventricularly [25]. Injection into the SNpc [26] or MBF leads to massive death of DA neurons within a few days, whereas injection into the CPu leads to more uniform, partial degeneration that develops over 1–3 weeks [27]. Specifically, a lesion affecting the dorsomedial parts of the striatum (including the nucleus accumbens) has greater effects on locomotion and drug-induced turning behavior, whereas lesions affecting ventrolateral parts of the caudate nucleus and putamen have greater effects on movement initiation, sensorimotor function, and motor dexterity behavior [28,29,30]. Therefore, injection of the toxin into the SNpc and MBF may mimic the end-stage of PD, whereas injection into the CPu may be more likely to be associated with early onset of the disease.

#### 2.1.2. 1-Methyl-4-phenyl-1,2,3,6-tetrahydropyridine (MPTP)-Induced Model of PD

The neurotoxic properties of MPTP were accidentally discovered in the last quarter of the 20th century when an analog of the synthetic opioid 1-methyl-4-phenyl-4-propionpiperidine (MPPP) was introduced as a recreational drug with effects similar to those of heroin. After injection of MPPP synthesized in a home-set-up laboratory, a 23-year-old student experienced severe bradykinesia, which responded to levodopa [31]. Further investigation discovered that besides MPPP, MPTP was also present in the mixture as a byproduct of the reaction. MPTP is a protoxin that produces the toxic metabolite MPP+, which impairs mitochondrial function by inhibiting complex I [32,33]. After systemic injection, MPTP rapidly crosses the BBB and is oxidized by monoamine oxidase B (MOA-B) to 1-methyl-4-phenyl-2,3-dihydropyridinium (MPDP+). Since MPDP+ is an unstable molecule, it is spontaneously dismutated to MPP+ [34,35,36], which is then released in the extracellular space and enters the DA neurons via the dopamine transporter (DAT) [37]. Once in the neurons, it can diffuse into mitochondria, where it inhibits complex I and increases the production of RO species [38,39]. This cascade triggers mitochondria-dependent apoptotic molecular pathways [40,41]. Interestingly, rats are generally more resistant to MPTP than mice, and only higher doses can make them Parkinsonian. The degree of striatum degeneration is relatively low compared to mice, and a higher dose is associated with a high mortality rate [42]. However, stereotaxic injection of 1-methyl-4-phenylpyridinium (MPP+), a toxic metabolite, has been used to induce a PD model in rats. Furthermore, it has been shown that intranasal administration of MPTP triggers DA degeneration and causes motor impairment in rats [43].

Symptoms and disease progression depend on several factors, such as the regimen of the MPTP administration [44], route of administration [41], sex, age, and weight, which are also important factors influencing sensitivity to MPTP and reproducibility of the lesion [45]. The acute regimen was developed by Jackoson-Lewis and Przedborski. It involved four injections at a dose of 20 mg/kg 2 h apart, resulting in a loss of 70–90% of dopaminergic neurons for at least 7 days after MPTP [44]. The sub-acute regimen was developed by Tatton and Kish and includes one i.p. injection of 30 mg/kg daily for five days, which leads to a loss of 40–50% of neurons and keeps the lesion stable for 21 days [46].

Regarding motor impairments, there are large behavioral differences among different outbred/inbred strains, and often these differences exist within subtypes of a given strain [47,48]. Specifically, the mice appear completely normal after recovery from the injections, meaning that very sophisticated testing is required to determine the presence and degree of the lesion [49,50], and some motor deficits are transient [51]. Similarly, both microglia and astrocytes show activation following MPTP intoxication, with microglia typically showing biphasic activation, whereas astrocytes tend to show a bell-shaped profile [52].

#### 2.1.3. Advantages and Disadvantages of Toxin-Induced Experimental PD

One of the main advantages of the 6-OHDA model is that the extent of neurodegeneration depends on the site of the lesion, the dose/amount of toxin, and the volume of microinjection, as well as the number of microinjections administered, which means that one can model the degree of neurodegeneration with relatively high accuracy and reproducibility. Another advantage is that this model is most comparable in behavior to the human disease and offers the possibility of tracking the progression of DA neurodegeneration and the effects of other neurotransmitter systems in PD pathogenesis of motor symptoms, as well as the prodromal phase with non-motor symptoms of PD [16,53,54]. Furthermore, 6-OHDA toxin is commonly used and validated for examination of levodopa-induced dyskinesia [55]. Another advantage of the unilateral 6-OHDA model is that the contralateral side can serve as an internal control. However, it should be noted that interhemispheric compensation may occur, and even DA levels on the contralateral side have been shown to increase after a 6-OHDA lesion [56]. The main disadvantage of the 6-OHDA model is its relatively rapid development and the absence of real progression of the pathology, with transient and low-grade neuroinflammation [57,58]. Another pitfall is the fact that this model is not suitable for examination of the role of immune cells in PD development and progression due to the nature of the induction. Finally, LB/α-syn accumulation is not present in this model [59].

The MPTP model has shown great value in studying molecular pathways involved in PD, as well as probing different therapeutical approaches. The relevance of this model is reflected in mitochondrial dysfunction, which is also observed in PD patients [60]. Additionally, it demonstrates strong face validity; for instance, MPTP intoxication leads to severe bradykinesia, which can be effectively treated with levodopa [31]. Another advantage of the MPTP model is that it allows researchers to work with genetically modified mice. Due to the nature of the application, this model allows for the study of the role of peripheral immune cells in the development and progression of parkinsonism [61] and changes in the gut–brain axis. Interestingly, peripheral immune infiltration is also a likely player in the neurodegenerative/neuroinflammatory process, as infiltration of CD8+, CD4+, and monocytes in the nigrostriatal system has been observed in MPTP intoxication [62,63,64]. However, one of the major drawbacks of the MPTP model, apart from the large variability described above, is the fact that the mice recover after a certain amount of time [65]. Furthermore, based on this recovery, it has been hypothesized that MPTP may not lead to cell death but rather to a decrease in the DAT or simple tyrosine hydroxylase (TH) impairment [66,67]. However, the progressive nature combined with the inflammatory component makes this model a suitable candidate to address one of these aspects in the treatment of PD [68,69].

In summary, the 6-OHDA model provides high accuracy and reproducibility for modeling the extent of neurodegeneration in PD. In contrast, the MPTP model, despite its recovery-related limitations, provides valuable insights into molecular pathways and immune cell involvement, making both models indispensable for various aspects of PD research.

### 2.2. Genetic Models of PD

Although the exact causes underlying PD are not yet fully understood, it is known that genetic variants linked to PD occur at different frequencies and lead to different levels of risk. At one end of the spectrum, certain rare mutations in single genes, such as SNCA (encoding α-syn protein) and PARK7 (encoding the protein DJ-1), are sufficient to cause the disease. Conversely, genome-wide association studies (GWASs) have uncovered numerous common genetic variants, each of which contributes only slightly to PD risk. Within this spectrum, there exist variants that are rare but not uncommon and have intermediate risk, such as those found in the GBA (glucocerebrosidase) and LRRK2 (leucine-rich repeat kinase 2) genes [70]. Among the genetic models used in research to mimic PD pathology in mice, transgenic mice overexpressing α-syn and LRRK2 gene mutation models stand out. Regarding genetic models, SNCA transgenic models do not show extra-nigral pathology, whereas LRRK mice show a slight increase in serotonin levels in the prefrontal cortex and a decrease in olfactory bulb dopaminergic neurons and locus coeruleus noradrenergic neurons at 24 months of age, and GBA mutant mice develop cholinergic dysregulation in the hippocampus [71]. These models provide insights into underlying mechanisms that contribute to disease development and potential treatments (Figure 1) [72,73].

### 2.3. Agrochemical-Induced Models of PD

Exposure to pesticides, particularly rotenone, paraquat (PQ), and maneb, has been linked to an increased risk of PD. Rotenone crosses the BBB and inhibits mitochondrial complex I (MCI) and proteasomal activity. This action triggers selective degeneration of DA neurons in the SNpc, leading to motor impairment and the formation of α-syn inclusions in rodents. PQ, a commonly used herbicide, might not directly penetrate the BBB. However, once it is metabolized in microglia to the monovalent cation PQ(+), it could enter DA neurons via the DAT. In rodents, PQ induces partial degeneration of DA neurons in the SNpc, transiently increases α-syn protein levels, and triggers oxidative/nitrosative stress, neuroinflammation, and microglial activation [74]. Thus, these models are useful for studying the role of neuroinflammation induced by pesticides in PD.

The various experimental models and their main characteristics are presented in Figure 1.

## 3. Peripheral Immunity in the Pathogenesis of PD

### 3.1. Adaptive Immunity

The adaptive IS achieves specificity through unique T- and B-cell receptors (TCRs and BCRs) generated by genomic recombination, allowing recognition of diverse pathogens [75,76]. Memory development in T and B cells enables swift responses upon re-encountering pathogens [77]. Infection-related immune responses have been linked to PD, demonstrating associations with influenza, toxoplasmosis, and Epstein–Barr virus (EBV) infections [78,79,80]. Studies in mice infected with the H5N1 influenza virus or studies with Japanese encephalitis virus (JEV) infection in humans indicate that inflammation-induced adaptive immune responses might contribute to neurological disorders associated with PD [81,82].

Numerous viruses are known to study or contribute to the development of parkinsonism and PD. Since the description of encephalitis lethargica after the Spanish flu in 1918, the link between the two diseases has been debated. While some viral infections are associated with an increased risk of PD, others appear to be directly linked to the manifestation of parkinsonism [83]. Viruses that possess neurotropic properties can cause direct damage to the nigrostriatal pathway as they enter the central nervous system (CNS) via three potential entry routes: 1. peripheral nerves; 2. BBB; and 3. blood–cerebrospinal fluid barrier. In addition, viral infections can indirectly affect the nigrostriatal signaling pathway by triggering inflammatory, vascular, and/or hypoxic injury [83].

Post-infectious parkinsonism is thought to arise as a result of pathogen-induced autoimmunity [83]. Viruses can trigger autoimmune responses through various mechanisms, such as “molecular mimicry, bystander activation and epitope spreading, with or without viral persistence” [83]. Molecular mimicry is a situation in which structural similarities between “viral and host antigens trigger T- or B-cell responses targeting both host and autoantigens” [84]; this can be observed in cases such as herpes simplex virus 1 (HSV-1) and Epstein–Barr virus (EBV) [79,85,86]. Routes of entry of HIV into the brain include the “Trojan horse” model (virus migration across the BBB by infected monocytes or lymphocytes) or the free virion model (by infected endothelial cells) [87]. Following entry into the brain, HIV has been shown to infect astrocytes, microglia [88,89], and neurons [90,91]. It is assumed that the infection of monocytes and macrophages by HIV leads to the production of neurotoxins [83].

#### 3.1.1. T Lymphocytes in PD

According to current evidence, the association of naive and memory T cells with PD suggests that these lymphocytes may play a role in disease initiation and propagation [92,93].

In animal models of PD, such as the acute MPTP neurotoxin model, CD8^+^ T cells are more present than CD4^+^ T cells [64]. The loss of DA neurons and behavioral problems resulting from both chronic and acute administration of MPTP are reduced in RAG2 (recombination activating gene 2) knockout (KO) mice (that lack both T and B cells). Mice with a global absence of αβ-T cells due to the KO of the T-cell receptor β-chain and mice lacking CD4^+^ T cells (CD4−/−) are both resistant to the effects of MPTP [94,95]. In the AAV (adeno-associated vector)-human-α-syn mouse model, B and T lymphocytes continue to infiltrate the SN even after microglial activation peaks, suggesting that adaptive immune cells contribute to inflammation [96]. These studies emphasize the role of the adaptive immune response in influencing neurodegeneration in PD models. T-cell responses to modified α-syn or other DA neuron antigens may initiate an immune response leading to neuronal death, depending on how these antigens are presented by innate immune cells. Identifying specific antigens that trigger T-cell responses in animal models and humans will be crucial for understanding the role of the adaptive IS in PD pathogenesis [97].

#### 3.1.2. Antibodies and B Lymphocytes in PD

In the adaptive IS, B lymphocytes produce antibodies crucial for humoral immunity [98]. Antibodies targeting CNS proteins in PD patients have been observed to exert effects in distant regions [99,100], but infiltration of antibody-producing B lymphocytes into the CNS has not been reported in PD [94,101,102]. In animal models of PD, experiments involving the direct injection of immunoglobulins from PD patients into the SN of rats have shown significant effects [103]. These injections resulted in inflammation and activated immune responses [104]. Also, studies using mouse antibodies against α-syn have shown that microglial cells can prevent the transfer of antibody-bound α-syn from neurons to astrocytes [105]. These models suggested a role for humoral immunity and potential BBB breakdown in contributing to chronic neuroinflammation in PD [96].

In summary, given the persistent inflammatory features of PD, it is thought that the humoral immune response may have a major impact on the progression, with T-cell immunity possibly having a greater impact in the early stages of the disease. The involvement of both naïve and memory T cells in the development and spread of PD is also a potential factor that needs to be considered.

## 4. Central Immunity

Neuroinflammation is considered a key feature in PD patients and in animal models [106]. In PD, there appears to be evidence of inflammation occurring both centrally and peripherally [107]. Traditionally, the CNS is considered immune-privileged due to the BBB that separates it from the peripheral IS. Under physiological conditions, microglia and astroglia actively maintain CNS homeostasis by releasing neurotrophic factors, regulating synaptic glutamate, and contributing to synaptic remodeling [107,108].

Neuroinflammation is present in most animal models of PD, although a single animal model does not replicate all the pathological features of the human disease. A recent study utilizing the reserpine (an irreversible inhibitor of the vesicular monoamine transporter 2)-induced rat model of progressive PD revealed a shift in microglial phenotypes from proinflammatory to anti-inflammatory 20 days after the last reserpine injection. This shift was observed in brain regions implicated in the pathophysiology of PD, highlighting a dynamic change in microglial responses depending on the disease stage [14].

Activation of microglia and alteration of inflammatory signals were detected in both mice and non-human primates (NHPs) after MPTP treatment, while astrogliosis and microgliosis in the SN and striatum of 6-OHDA-treated mice and rotenone-treated rats are well documented. LPS injected into the SN to model PD increases proinflammatory cytokine levels. In addition, an increase in microglial numbers and proinflammatory cytokines is observed in the striatum after AAV-α-syn injection, even before DA neuron death occurs. In the α-syn preformed fibril (PFF) mouse model, there was a change in immune cell content in the periphery, microgliosis in several brain regions, and the presence of reactive astrocytes in the SN [71]. Despite the advances in understanding the role of neuroinflammation in PD animal models, further studies are needed, especially in the genetic models of PD.

### 4.1. PD and Resident Immune Cells

Activation of microglia and subsequent neuroinflammation are associated with the degeneration of DA neurons, particularly in the SN pars compacta (SNpc) [101].

Microglia play a role in the clearance of misfolded α-syn aggregates in PD, but when overactivated, they can produce proinflammatory cytokines and ROS and damage DA neurons [109,110]. Under normal conditions, microglia, which are long-lived macrophages in the CNS, remain in a dormant state and perform immune surveillance through their highly mobile processes [111]. They can react quickly to pathological changes in the CNS. Microglia, which act as phagocytes, remove invaders and debris in the brain. When exposed to specific stimuli, they can enter an activated state, alter gene expression, and exert inflammatory functions. Technological advances, particularly single-cell and nuclear RNA sequencing, have revealed different states of microglia, improving our understanding of their complexity and unveiling regional variations within the CNS [112,113].

Animal models have played a crucial role in understanding the involvement of microglia in the pathology of PD. Different neurotoxins, such as MPTP and 6-OHDA, lead to varying degrees of microglial activation in affected areas. In addition, researchers have used proinflammatory stimuli such as LPS, α-syn, and inflammatory cytokines to activate microglia, leading to toxicity in DA neurons. While all these models result in some degree of activated microglia in the vicinity of damaged DA neurons, they differ in the timing of microglial activation in response to the toxic stimulus. Furthermore, the sequence of events in DA areas varies, with microglia activation occurring either before neuronal damage or as a secondary event, depending on the specific animal model being studied [114,115].

Astrocytes play an important role in supporting neurons, contributing to metabolic balance, synapse formation, and the maintenance of brain structure and the BBB [116,117]. While they have a protective function by isolating and degrading extracellular α-syn, elevated levels of α-syn can induce inflammatory responses in astrocytes, potentially worsening synucleinopathy-related conditions [118]. Studies on reactive astrocytes reveal a dual nature where A1 astrocytes contribute to neurodegenerative diseases with a proinflammatory and neurotoxic profile, while A2 astrocytes display a neuroprotective function. Microglial activation can convert astrocytes into the neurotoxic A1 phenotype, but blocking this conversion with a GLP1R agonist (NLY01) shows neuroprotective effects in vivo against DA neuronal loss and behavioral deficits in a mouse model of sporadic PD [119]. The researchers isolated pure populations of resting astrocytes using immunopanning and cultured them. Through a microfluidic assay, they explored the impact of various molecules on gene expression, revealing that A1 astrocyte activation is triggered by the combined effects of interleukin (IL)-1α, TNF, and complement component 1q (C1q) released by microglia in response to LPS stimulation. Once activated, A1 astrocytes contribute to neuroinflammation-mediated neurotoxicity and exacerbate the progression of neurodegenerative diseases [120].

Oligodendrocytes, responsible for myelin production, are under investigation for their role in PD, with studies suggesting involvement in neuroinflammation and immune response modulation. In a recent study using an in vivo mouse model simulating inflammation-induced white matter injury in preterm-born pups, researchers observed distinct responses to neuroinflammation between immature oligodendrocytes and oligodendrocyte progenitor cells, particularly in IL-1β-treated animals [121,122,123]. The neurovascular unit (NVU), which is composed of neurons, glial cells (astrocytes, oligodendrocytes, and microglia), and vascular cells (endothelial cells and pericytes), is essential for the establishment of tight junctions. It plays a crucial role in maintaining key functions, such as the BBB, ion balance, and nutrient transport [124,125,126]. Brain endothelial cells, with specialized junctions and high connexin expression, are particularly important for the properties of the BBB and limiting paracellular permeability [127].

Pericytes, located in the walls of microvessels, play a crucial role by interacting with endothelial cells, neurons, glial cells, and perivascular macrophages [128,129]. Recent studies highlight their involvement in neuroinflammatory and neurodegenerative diseases [130], with pericytes producing pleiotrophin, a vital neurotrophic factor for neuronal survival. Consistent findings from single-cell RNA sequencing revealed a higher expression of pleiotrophin in pericytes compared to low expression in astrocytes, oligodendrocytes, and endothelial cells. Removal of pericytes in mice resulted in the immediate breakdown of the BBB, loss of blood flow, and rapid loss of neurons, highlighting the importance of pericyte-derived pleiotrophin [131]. In addition, pericytes activated by α-syn release proinflammatory molecules, indicating their role as the first step in vascular changes and pathological signaling events within the NVU in PD [132]. This suggests that pericyte activation could be the initial step in vascular changes and several pathological signaling events within the NVU in PD.

Overall, the IS and its interactions with various cell types in the CNS play a significant role in the pathogenesis of PD. Further understanding of these immune-mediated mechanisms may lead to potential therapeutic targets for PD treatment.

The role of peripheral and central immunity in the pathogenesis of PD is shown schematically in Figure 2.

### 4.2. Tissue Infiltration

Various animal models are used to study the infiltration of the CNS by immune cells and the role of different cellular products such as α-syn. In a mouse PD model, overexpression of pathogenic A53T-α-syn (haSyn) by injection of an AAV leads to T-cell infiltration. Research with T-cell- and/or B-cell-deficient mice, along with a novel haSyn neuronal cell culture and immune cell assay, confirms that pathogenic α-syn peptide-specific T-cell responses lead to DA neurodegeneration and contribute to PD-like pathology [133,134]. In haSyn PD mice, ~80% of the observed T cells were located in the SN parenchyma [134].

Activated microglia and astrocytes produce a variety of proinflammatory cytokines, including TNF-α, IL-1β, and IFN-γ, creating an environment that mediates cell death [135]. Moreover, activated microglial cells, peripheral myeloid-mediated proinflammatory innate immune responses, and CNS neurotoxic adaptive immune activity have been implicated in the pathogenesis of 6-OHDA-induced PD. Brain imaging was performed using PET/CT imaging and ex vivo autoradiography. Immunohistochemistry was used to observe myeloid cell activation and DA cell death, and quantitative polymerase chain reaction and flow cytometry methods were used for target levels in the brain [136]. In particular, whereas neuroinflammation induced by α-syn has been shown to exacerbate neurodegeneration, the role of CNS resident macrophages in this process remains unclear. Border-associated macrophages (BAMs), a subset of macrophages resident within the CNS, have been implicated in initiating the CD4 T-cell response and, therefore, mediating α-syn-associated neuroinflammation. Furthermore, brightfield and fluorescent imaging and quantification, confocal imaging, immunohistochemistry, and isolation of mononuclear cells demonstrated that the presentation of major histocompatibility complex class II (MHCII) antigens on microglia does not affect neuroinflammation. In addition, BAMs were identified in proximity to T cells in postmortem PD brains. These results suggest that BAMs play a role in regulating the α-syn-mediated neuroinflammatory response in the pathogenesis of PD [137].

In the 6-OHDA-induced PD model, increased microglial activation was found to precede the loss of DA neurons, suggesting that phagocytic microglia may prematurely engulf degenerating neurons [138]. 6-OHDA destroys nigrostriatal dopaminergic neurons, causing motor and biochemical dysfunctions in PD [58,139]. In the microglia and macrophage ratios taken from 6-OHDA lesioned rats, decreases in the ratio of CD80/86+ cells were observed, while the increased fraction of CD206+ cells was noted [58]. However, in the MPTP model, inflammatory markers in the SN, such as HLA-DR-positive reactive microglia and infiltrations of T lymphocytes, including CD8+T cells and CD4+T cells, were observed by immunohistochemistry and electron microscopy [94,135,140]. In the MPTP mouse model, CD8^+^ and CD4^+^ T cells, rather than B cells, infiltrate the brain during neuronal degeneration. The damaging activity of the infiltrating CD4^+^ T cells is mediated via the Fas/FasL pathway and not via IFNγ production [94]. Further studies have shown that peripheral immune infiltration is also a likely player in the neurodegenerative/neuroinflammatory process, as infiltration of CD8+, CD4+, and monocytes was observed in the nigrostriatal system after MPTP treatment. To evaluate monocyte infiltration in the SN, laser microdissection-guided chemokine RNA profiling was evaluated by combining immunohistochemistry and chemokine receptors CCR2-green fluorescent protein (GFP) (CCR2-GFP) methods [62,63].

Microglial activation and increased expression of inflammation-related proteins such as TNF-α, IL-1β, and NFκB were detected in the SN of PQ-treated rats [141,142,143,144]. In addition, rotenone leads to the activation of microglia in animal models. Furthermore, rotenone leads to microtubule instability and progressive death of DA neurons, along with the formation of α-syn aggregates in rats [145]. Despite these consistent results, further studies on strategies to suppress microglial activation are needed to clarify the exact role of microglia in rotenone-induced degeneration [146].

The basic interactions between PD neuroinflammation and peripheral inflammation have not yet been clarified. In a PD model, data obtained by PET/CT, ex vivo autoradiography, immunohistochemistry, quantitative polymerase chain reaction, and flow cytometry were analyzed: PD is associated with abnormal innate immune responses, including infiltration of peripheral myeloid cells into the CNS [136]. As mentioned before, innate immune responses triggered by microglia can cause neuronal death and disease progression. In addition, T cells infiltrate the brains of PD patients and are involved in adaptive immune responses. Interestingly, α-syn is associated with both innate and adaptive immune responses by directly interacting with microglia and T cells [147]. There is evidence that the innate immune system plays a role in DA cell death. The measurement of striatal dopamine, electron microscopy images, and immunohistochemistry analysis have collectively contributed to the conclusion that T-cell-mediated DA toxicity is primarily influenced by CD4+ T cells. A more detailed characterization of CD4^+^ T-cell subsets will certainly improve our understanding of the role of adaptive IS in various neurodegenerative disorders [94]. Antigen specificity is important for neuroimmunological diseases. α-synuclein is thought to be the cause of PD. However, the lack of a causal link between α-syn responses and DA neurodegeneration has attracted attention. Methods such as immunohistochemical stainings and cell quantification, double immunofluorescence, fluorescence-activated cell sorting (FACS) and cytokine assay, striatal DAT autoradiography, high-performance liquid chromatography (HPLC), immunocytochemistry, and measurement of fluorescence intensity are used. It has been emphasized that targeting α-syn and α-syn-specific T cells could serve as a neuroprotective strategy in the immunomodulation of PD [134].

So far, the methods to study innate immune activation in PD are limited. Understanding the interactions between neuroinflammation and peripheral inflammation remains an important research issue in PD.

### 4.3. Model to Assess Tissue Infiltration

Most of the PD experimental animal models are performed on rodents, including mice and rats. One of the first studies on mice used the MPTP administration model [148]. In those experiments, irradiated mice were reconstituted with bone marrow from GFP+ mice, and the infiltration of GFP+ cells in midbrain tissue sections was detected by immunofluorescence. Immunohistochemistry showed no infiltration of B lymphocytes into the SN after MPTP, but most infiltrating lymphocytes were CD25+, and the density of CD8+ lymphocytes was higher than that of CD4+ lymphocytes. A later study also showed increased MPTP-induced infiltration of peripheral CD4+- and CD8+-positive T cells in the SN and spinal cord compared to controls, based on an immunofluorescence approach [149]. Furthermore, in vitro-generated brain Treg (regulatory T cells)-like cells (iB-Treg cells) induced via co-culture with astrocytes under several conditions were injected into MPTP-treated mice, and flow cytometry was used to investigate the presence of brain-infiltrating T and Treg cells, corroborating the sensitivity of immune cells to this toxic agent [150]. Kozina et al., in 2018, utilized transgenic mice that carried the two most common pathogenic *LRRK2* mutations and induced innate response by LPS injection. Immunohistochemistry on brain tissue, along with flow cytometry on single-cell brain suspensions, revealed no infiltration of T cells or monocytes [151]. Another PD mouse model in which an AAV-overexpressing human mutated A53T-α-syn was stereotaxically injected into the SN showed that a-syn overexpression induces infiltration of immune cells, particularly CD4+ and CD8+ T cells, which was demonstrated by flow cytometry [134]. In addition, Lucot et al. used translocator protein positron emission tomography (TSPO PET) and in vivo PET imaging of TREM 1 (triggering receptor expressed on myeloid cells 1) in a 6-OHDA mouse model of PD. They demonstrated neuroinflammation, specifically the infiltration of myeloid cells, using flow cytometry [136].

In addition, liposomes targeting the CD163 receptor were loaded with glucocorticoids and injected peripherally in a 6-OHDA-PD rat model. The treated rats showed infiltration of CD163+ macrophages, particularly into the area of neurodegeneration [152]. In 2006, a study on MPTP-treated monkeys showed infiltration of LFA-1 (lymphocyte function-associated antigen 1)-positive leukocytes into the brain [153]. The transplantation of autograft adrenal or allograft fetal mesencephalic tissues into the CNS of monkeys was developed as a therapeutic approach for NHP PD. Bakay et al. tried to evaluate the potential pathology of transplantation at the graft site using immunohistological approaches [154]. They demonstrated the presence of mononuclear cell infiltrates, but the transplantation protocol did not induce detectable donor-specific sensitization nor nonspecific immunosuppression [154]. On the other hand, in a study on zebrafish and tissue regenerative capacity, Zwi et al. showed that the DA agonist pramipexole, a drug currently approved for treating PD, specifically enhanced Treg recruitment after injury [155]. For this purpose, FOXp3a+ Tregs were visualized by epifluorescence in the regenerating tail region. Overall, fluorescent-based methods, such as flow cytometry or microscopy, are the most widely accepted approaches for visualizing immune cell infiltration in the brain in experimental animal models of PD. These studies have shown increased infiltration of both CD4+ helper and CD8+ cytotoxic T lymphocytes into the brain in PD animal models. Interestingly, the infiltration of CD163+ macrophages in the inflamed brain aligns with findings from human studies regarding the role of peripheral monocytes in PD [156].

## 5. Other Models to Study Immune Dysfunction in PD

### 5.1. Zebrafish

The zebrafish has emerged as a disease model, particularly due to its transparency at the larval stage, its high content of orthologs of human disease genes, its genetic reproducibility, and the affordable cost of housing. The hematopoietic system in zebrafish resembles that of other vertebrates; however, the development of the adaptive IS does not occur until approximately 20 days post-fertilization (dpf). As in mammals, microglia are the resident immune cells of the zebrafish brain. At three to five dpf, zebrafish larvae develop a microglial signature and express a significant number of genes that they share with mouse and human microglia [157]. At this early stage of development, translucent zebrafish larvae possess DA neurons, which can be targeted by the administration of neurotoxins directly into the media [158], thus making them a simple and affordable model for studying the link between loss of DA cells and neuroinflammation.

The invasion of peripheral macrophages into the zebrafish brain following an injury is a subject of debate [159,160]. Neither, to our knowledge, has it been shown that T cells or B cells enter the zebrafish brain. However, using a transgenic zebrafish line with cyan fluorescent protein-labeled regulatory T cells (zTreg) that could be ablated using the pro-drug metronidazole (Mtz), it was demonstrated that zTreg translocated to injured sites after Mtz washout. This translocation mediated the regeneration process in the retina and spinal cord, as confirmed by both microscopy and flow cytometry [161].

As in rodents, administration of 6-OHDA into adult zebrafish leads to a loss of DA neurons. Nevertheless, in contrast to mammalian models, zebrafish are capable of regenerating subpopulations of these neurons [162,163]. Thus, they provide a model to elucidate neuronal regenerating mechanisms that are lacking in mammalian models. The replacement of DA neurons in the zebrafish was dependent on IS activation involving increased proinflammatory cytokines (IL-1β and TNFα) and microglial activation as measured by qRT-PCR and immunohistochemistry [163]. Using a combination of transgenesis and single-cell analysis, Oosterhof et al. showed that neuronal ablation primarily induced a proliferating response pattern in microglia [160].

Stable gene-based zebrafish KO models have been established for the PD-associated genes *park7* and *pink1* [164,165]. In the *park7* KO, a dysregulation in proteins involved in inflammation and mitochondrial metabolism was observed before altered behavior and downregulation of TH [165,166]. Loss-of-function models are relatively easy to establish in zebrafish using CRISPR/Cas9-based technology. As more than 90 genetic risk factors have been identified for sporadic PD [167], zebrafish appear to be an excellent model for elucidating the role of inflammation associated with these PD risk factors. For example, the GTP cyclohydrolase (*gch1*^−/−^) zebrafish mutants exhibit a reduced amount of TH, although they do not show loss of DA neurons. On the other hand, RNAseq of larvae brains revealed a significant upregulation of transcripts involved in innate immune response and microglial activation [168].

### 5.2. Cellular Models

The study of the role of the IS in PD is a constantly evolving field [5]. Cellular models are an indispensable tool for understanding the intricate interplay between the IS and the pathological progression of this disease [169]. Below is a selection of the cellular models used in PD and IS research.

#### 5.2.1. Microglial Cell Cultures

Microglial cell cultures have been used to understand how these cells respond to inflammation and neuronal damage in PD, and they facilitate the study of microglial response to genetic or toxicological backgrounds associated with PD under controlled conditions (temperature, concentration of substances, etc.) [170,171]. Therefore, microglial cell cultures allow us to gain insights into the response of microglia in PD, with less variability than when studying in vivo models or PD patients.

A major disadvantage of using primary microglial cultures is that cell cultures do not consistently replicate the behavior of cells in a whole brain and, therefore, cannot replicate the complex cellular environment and interactions with other cell types.

As obtaining primary microglial cultures from patients is highly invasive, microglial cells derived from induced pluripotent stem cells (iPSCs) obtained from human fibroblasts are currently being studied [172,173]. However, it is not known whether fibroblast-derived microglial cells behave in the same way as brain-resident microglial cells. Furthermore, inter-individual variability between patient samples introduces an additional level of fluctuation, which is a further limitation.

#### 5.2.2. Culture of Peripheral Immune Cells: Monocytes–Lymphocytes

Culturing peripheral immune cells, including monocytes and lymphocytes, is a common procedure in immunology and cell biology research. These cultures allow researchers to study the behavior, function, and response of these immune cells under controlled laboratory conditions [174,175].

The immune cells of mice can be isolated from the brain, peripheral blood, spleen, or bone marrow by immunomagnetic separation and/or cell sorting, and their phenotypes can be studied. In addition, the transplantation of immune cells in these animal models could provide information on the neurodegenerative or neuroprotective effects of different immune cell populations. In this context, it is important to consider the characteristics and limitations of the individual experimental models.

To explore the immune cell responses associated with PD, the culture medium could be supplemented with proinflammatory cytokines (e.g., interleukin-1β, IL-1β) or neurotoxins (such as LPS) to induce cell activation. The subsequent functional assays could then be used as tools to assess various physiological parameters, including cytokine production, oxidative stress, monocyte phagocytic activity, and T-cell activation in lymphocytes.

#### 5.2.3. Co-Culture Models

Cellular co-culture models are valuable tools in PD research as they allow scientists to study the interactions between immune cells and neurons in the context of the disease [176]. Some approaches to generate co-culture models that focus on the role of IS in PD are described below.

Microglial cells and/or astrocytes from animal models of PD can be co-cultured with primary DA neurons or other relevant neuronal cell types to study the effects of microglial activation on neuronal survival and function. These models facilitate the study of the interplay between astrocytes, microglia, and neurons in the context of inflammation and oxidative stress associated with PD and provide valuable insights into the involvement of astrocyte-mediated immune responses [177,178,179,180].

Some animal models of PD show infiltration of peripheral immune cells into the CNS [94,96]. Co-culture models of isolated peripheral immune cells (e.g., T cells or monocytes) from these PD models with brain-derived cells such as neurons or microglia can be generated and evaluated. This approach could help in the investigation of the effects of immune cell infiltration on neuronal health and inflammation in the CNS [181,182].

Co-culture of iPSC-derived neurons from animal models or PD patients, together with microglia or macrophages [183] or in conjunction with immune cells such as peripheral blood lymphocytes [184], is a valuable approach to gaining deeper insights into the intricate interplay between different cell types and the role of IS in PD [185,186].

#### 5.2.4. Ex Vivo Brain Slice Cultures with Immune Cells

Brain slice cultures from the ventral mesencephalon or striatum of PD animal models allow us to study the interactions between different cell types. In addition, immune cells can be introduced directly into the slices to study the interactions between immune cells within the brain tissue and their effects on neuronal health and inflammation in a more physiological environment [182,187].

#### 5.2.5. Organoids

Brain organoids are generated from iPSCs or embryonic stem cells (ESCs) to model the structure and function of the whole brain. Brain organoids can contain neurons, astrocytes, and microglia-like cells and allow researchers to mimic the complexity of brain and cellular interactions [188,189]. Cells from organoid-derived animal models or PD patients may have relevant genetic or physiological changes associated with this disease [190]. Medial brain organoids do not contain some types of glial or immune cells, so it is necessary to culture them with additional non-neuronal cell types. Immune cells from PD animal models could be introduced into the organoids to simulate the immune response observed in PD [184,191]. Exposure to inflammatory stimuli and substances associated with PD enables researchers to study the interaction between immune cells and neurons and assess their role in neuroinflammation and neurodegeneration [192].

The advantage of co-culture models is that they provide a platform for studying the dynamic and complex interactions between immune cells and neuronal cells, helping researchers to elucidate the role of IS in PD pathogenesis, neuroinflammation, and potential therapeutic interventions. All of these experiments are conducted in a controlled and reproducible experimental setting, which facilitates the manipulation of specific variables and the assessment of their effects on immune responses and neuronal well-being. They can also be used to screen drugs and test new immunomodulatory treatments for PD [193,194].

Culture models may not accurately represent the complexity of PD progression and can only mimic long-term effects to a limited extent. Animal model cells may not accurately reflect all aspects of PD pathology, making translation of results difficult. Technical difficulties in isolating cells also contribute to experimental variability. Therefore, validation of these models by more complex in vivo systems or human studies is crucial for clinical relevance.

In Table 1, a summary of the advantages and disadvantages of the cellular models for studying the interaction between the IS and PD are shown.

In summary, culture models provide a valuable platform for investigating the role of IS in PD. However, researchers should be aware of their limitations and carefully consider the advantages and disadvantages when designing experiments and interpreting results.

## 6. BBB Alterations in PD and the Immune Response

During aging, in neurodegenerative diseases such as PD and in immune disorders, the integrity of the BBB can be compromised, leading to increased permeability to materials and cells from the bloodstream [195]. Using a human wild-type α-syn gene that is fused to the green fluorescent protein (GFP) gene and overexpressed under the mouse α-syn promoter in mice, Elabi et al. developed a transgenic animal model that showed α-syn accumulates and PD-like symptoms with age, mimicking the human disease. They then analyzed the longitudinal changes associated with the BBB. They found a very early appearance of leakage in the BBB and activation of pericytes that may contribute to the pathogenesis of the disease [196]. In another study, a PD mouse model that overexpresses human wild-type α-syn was analyzed, and several early BBB changes were detected. These included changes in striatal capillaries, decreased vascular density, alterations in AQP-4 coverage, reduced P-glycoprotein (P-gp) expression, increased LRP1 expression, deposition of pS129-α-syn, elevated expression of VCAM-1 and MMP-3, and an early reduction in the tight junction protein occludin [197]. These changes lead to a pronounced leakiness of the BBB when a toxic insult occurs, such as LPS injection.

From the astrocyte perspective, there are also data showing early astrocyte activation and BBB changes [198]. Exposure to oligomeric α-syn has been shown to result in significant astrocyte activation and release of VEGFA (vascular endothelial growth factor A), which, in turn, results in BBB degradation. Furthermore, when astrocytes are activated by inflammatory stimuli, they may interact with endothelial cells, pericytes, and microglia. Therefore, they may be involved in the progression of PD and neurodegeneration [199].

The inflammatory mediators can also trigger and/or amplify immunological mechanisms. Serum-derived inflammatory mediators, such as TGF-beta, can interact with BBB cells, activating the inflammasome complex and pericytes. This activation leads to enhanced immune responses and disruption of the BBB [200], resulting in an increased number of immunological cells, such as T and B cells, in the brain parenchyma [94].

## 7. Glymphatic System and PD

In contrast to other organs, there is no structural system for lymphatic drainage in brain tissue. Lymphatic drainage is very important for the removal of metabolic waste and the regulation of immunological responses.

In 2012, Maiken Nedergaard and colleagues described the glymphatic system (GS), where cerebrospinal fluid (CSF) and BBB interact in the removal of waste products in brain tissue [201]. The name of this system is composed of the terms “glial cells” and “lymphatic system” and refers to a waste removal system that is primarily controlled by glial cells, particularly astroglia [202]. The glymphatic flow reaches interstitial space through Aquaporin-4 (AQP-4) water channels located in the end foot of the astrocytes [201].

GS plays an important role in the clearance of α-syn from the brain, whereas AQP-4 dysfunction accelerates the pathological deposition of α-syn, promoting the loss of DA neurons and accelerating PD-like symptoms [203]. These data were obtained by inducing a PD model through the injection of a vector expressing A53T-α-syn into the SN of AQP-4-deficient mice and controls, demonstrating that GS dysfunction leads to decreased clearance of α-syn. Correlated with the animal model, DTI (diffusion tensor imaging) in the brain was performed in PD patients and control subjects [204]. Using the DTI-ALPS (diffusion tensor image analysis along the perivascular space) index as an indicator of glymphatic circulation, the authors found a significant decrease in glymphatic circulation that correlated with the severity of motor symptoms [204]. Further studies on the role of the GS in PD pathogenesis were performed in the A53T transgenic PD model, showing that occlusion of cervical lymph nodes of the GS significantly aggravated α-syn pathology and led to dysfunctions of coordination and balance [205]. In line with this in vivo animal PD model, a study found that the condition of PD patients with a low DTI-ALPS index deteriorated more rapidly, and there was a positive correlation with autonomic dysfunction [206] as well as enlarged perivascular spaces also related to freezing of gait [207] and the stage of the disease in PD patients [208].

Research on the glymphatic pathway is new and promising. Studies in both animal models and PD patients are necessary to better understand the role of the GS in the pathophysiology of this disease.

## 8. Discussion and Conclusions

This review provides evidence that experimental animal models are valuable tools for studying the characteristic features and aspects of PD disease and offer a variety of approaches suitable to understanding the role of immune dysfunction in PD genesis and progression. The study of PD benefits from a wide range of available animal models, but this complicates the selection of the optimal model for a particular research purpose [11,71,209].

The pharmacological models of PD, induced by differential peripheral administration of reserpine and haloperidol, were the first to be developed and contributed significantly to the discovery of symptomatic drugs such as levodopa. Although the pharmacological models of PD are the simplest to create, they are transient. For more permanent effects, toxic animal models were developed with neurotoxins (6-OHDA and MPTP), pesticides (rotenone, paraquat, and permethrin), and endotoxins (lipopolysaccharide; LPS).

Toxin-induced models represent a cost-effective approach characterized by validity, accuracy, and high reproducibility for the investigation of different facets of PD. These models allow precise and graded control of DA neuron degeneration and provide a valuable platform for studying molecular signaling pathways, immune cell involvement, the gut–brain axis, and other important aspects of PD.

With the discovery of several forms of familiar PD, researchers have developed genetic animal models that attempt to replicate the disease through genetic mutation. There are at least 13 different α-syn (SNCA) mutation models alone, in addition to the mutations in leucine-rich repeat kinase 2 (LRRK2) and the lysosomal enzyme glucocerebrosidase (GBA) in mice, with clear links to both familiar and sporadic forms of PD. The SNCA model is associated with the pathology of PD and has been widely used to study spreading, toxicity, misfolding, and aggregation. It also led to the development of non-transgenic α-syn models by using AAVs to overexpress α-syn in the CNS of wild-type rodents and non-human primates, as well as the PFF model. All of these animal models of PD have different pathological features and are differentially compatible with the human disease [71].

Animal models are essential to study the role of BBB penetration by immune cells and their effects on CNS, especially in the well-known process of neuroinflammation. In this context, a wide range of fluorescent-based methods, such as flow cytometry or microscopy, are often used.

Additional approaches for preclinical studies of PD are cell cultures derived from animal models, which can provide valuable insights into the underlying pathophysiological mechanisms of this disease [71]. It is worth noting that the study of cellular models of IS in PD could benefit from the methodology already established in other neurodegenerative diseases where the role of the IS has been previously elucidated.

Considerable progress in the differentiation of distinct cell types from human fibroblast-derived iPSCs and the development of human organoids [184,210] has outpaced the development of corresponding cultures from animal models of PD. Nevertheless, we must not lose sight of the advantages of cell cultures from PD animal models, as they allow a reduction in the number of environmental and genetic variables and thus promote the generation of potentially more reproducible results.

In particular, specific animal models are employed to investigate the interplay between peripheral and central immunity in PD, a relationship that is increasingly recognized as pivotal for comprehending its complex mechanisms. Understanding the interplay between these ISs is crucial as it sheds light on potential triggers, disease progression, and the development of novel therapeutic interventions for PD. Animal models, in this case, offer the possibility to assess CNS infiltration by immune cells and the role played by these cells in neurodegeneration.

Animal models are essential to understand the key role of IS in neuroinflammation and neurodegeneration. The 6-OHDA-induced PD and MPTP models are frequently used to understand the role of microglial activation or to try to find markers of inflammation in the SN [94,135,140].

In addition, transgenic mouse models, such as the α-syn mouse model of PD [137], have changed the classical understanding of neuroinflammatory mechanisms in neurodegenerative diseases, showing the key role of border-associated macrophages (BAMs) in mediating α-syn-related neuroinflammation, due to their unique role as antigen-presenting cells required for initiation of a CD4 T-cell response.

However, it is important to keep in mind that all these diverse arrays of animal models in PD research, which replicate various aspects of the disease, require careful consideration when it comes to selecting the most appropriate animal model based on specific research needs and translating the results to what happens in humans. All the models mentioned in this review, and any others that we did not mention for the sake of brevity, as the term implies, are models of PD and cannot fully explain the complexity of the disease as it occurs in humans.

Table 2 shows some of the animal models and technical approaches used to study (especially visualize) the infiltration of immune cells into brain tissue. The wide range of animal models and methods, such as flow cytometry and the isolation of specific cell types, to investigate the role of the immune system in PD is offset by the limitation that the results are sometimes not reproducible.

Animal models and technical approaches to visualizing immune cell infiltration in brain tissue. As shown in the table, the majority of the methods used so far to visualize immune cells are based on fluorochrome-conjugated, lineage-specific antibody methodologies.

Therefore, we believe that in the future, the development of methodological guidelines for experiments involving both animal models and PD patients is crucial to ensure the validity and consistency of the results.

## Figures and Tables

**Figure 1 ijms-25-04330-f001:**
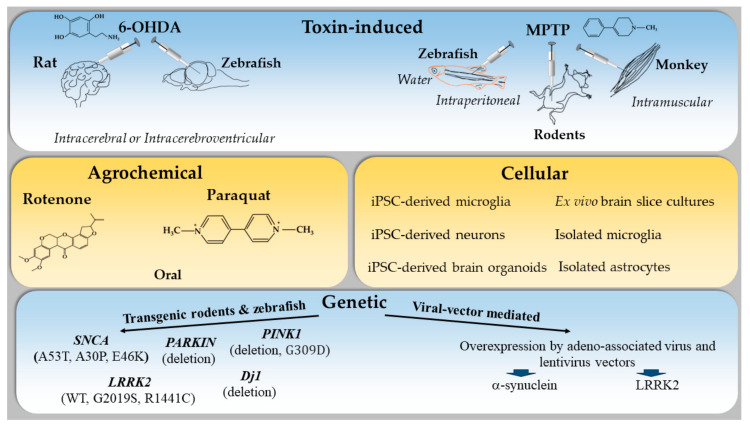
Schematic summary of the animal models of PD to study the immune system.

**Figure 2 ijms-25-04330-f002:**
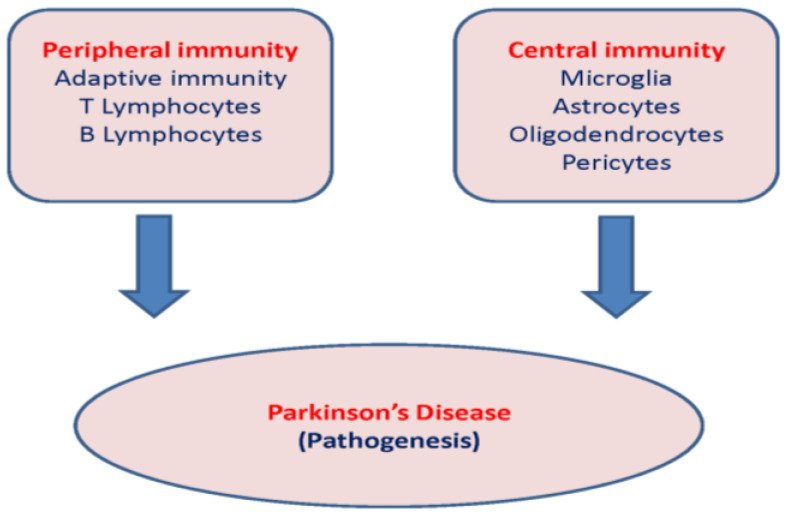
Peripheral and central immunity in the pathogenesis of PD.

**Table 1 ijms-25-04330-t001:** Cellular models: advantages and disadvantages.

Cell Culture Models	Advantages	Disadvantages
Primary microglial cultures	Study of the effect of genetic or toxic alterations associated with PD Controlled conditions, isolated effect	Does not replicate the complexity of the environment or the interaction with other cells of PD
IPSCs microglial-derived cells	Less invasive than primary cultures	It is unknown whether it replicates the phenotype of resident microglia
Culture of peripheral immune cells	The study of the phenotype of peripheral immune cells Transplantation from PD models to controls or vice versa (in vivo effects)	Require in vitro stimuli to induce cellular activation (e.g., LPS, oxidative stress, etc.)
Co-culture models:	They allow for the study of cellular interactions between CNS cells and IS cells under controlled conditions	The structural and environmental complexity is not replicated
Microglial + Astrocites + Neurons		
Peripheral IS cells + Brain cells		
IPSCs neurons + IS cells (microglial and macrophages)		
Ex vivo brain slice cultures + immune cells	Better replicate structural complexity and can be co-cultured with immune cells to study their interaction	Does not allow for the study of long-term effects
Organoids	Mimic the complexity of brain and cellular interactions Reproduces the genetic background of the animal model or the patient	Does not contain some types of glial or immune cells, needs co-culture with them Requires a long time for proper organoid maturation

**Table 2 ijms-25-04330-t002:** Technical approaches to studying immune cell infiltration in brain tissue.

Animal Model	Technical Approaches	Ref.
MPTP/mice	Immunofluorescence of GFP^+^ cells	Depboylu et al., 2012 [148]
MPTP/mice	Immunofluorescence	Samantaray et al., 2015 [149]
MPTP/mice	Flow cytometry	Yamamoto et al., 2022 [150]
LRKK2 transgenic mice	Immunohistochemistry and flow cytometry	Kozina et al., 2018 [151]
A53T-α-synuclein injection	Flow cytometry	Karikari et al., 2022 [134]
6-OHDA/mice	TSPO PET scan	Lucot et al., 2022 [136]
6-OHDA/rat	Immunofluorescence	Tentillier et al., 2022 [152]
MPTP/monkeys	Immunohistochemistry	Miklossy et al., 2006 [153]
Transplantation/monkeys	Immunohistology	Bakay et al., 1998 [119]
Zebrafish	Epifluorescence	Zwi et al., 2019 [155]

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
