# Peer review of "Experimental Models to Study Immune Dysfunction in the Pathogenesis of Parkinson’s Disease"

_ijms, 2024, doi:10.3390/ijms25084330_

Round 1

Reviewer 1 Report

Comments and Suggestions for Authors

I thank the authors for this interesting review. Unfortunately, it needs significant improvement. The authors should consider the relevance of each animal model to human’s Parkinson's disease. It is necessary to describe the correspondence between the processes occurring in humans and the processes in model animals.

The authors need to work on the consistency of presentation of the material. It may be worth highlighting two main parts: (1) Toxin-induced and genetic models (2) Methods for studying various disorders in PD (Neuroinflammation, immune dysfunction, Tissue infiltration and others).

The lack of illustrations is a significant drawback. Please add a figure that summarizes information about all PD models.

Specific Comments:

Please expand the introduction section. Please indicate the main mechanisms of development of PD in humans, on the basis of which animal models were created. The purpose of this review should also be stated.

Section 1.1. More detailed information about α-synuclein, Lewy bodies, non-dopaminergic pathways and their role in PD should be provided. Please add a picture.

Section 1.3. Why are 6-OHDA and MPTP used to create animal models? What is the mechanism of action of these substances?

Sections 1.3.1.1. and 1.3.1.2. Please describe the 6-OHDA-model and MPTP-model in more detail: symptoms of the disease, progression of the disease, MRI brain or brain slices, behavioral characteristics and more. The advantages and disadvantages should also be noted in these sections.

In section 1.3.1.3. the study of the role of immune cells in the development of PD is mentioned. What is known about their role in disease progression? Please add information about this in humans with PD and in animal models

Section 1.3.2. Please tell us in more detail about the mechanisms of PD occurrence with overexpressing α-syn and LRRK2 gene mutation. Please add information about experimental animals with these mutations: disease progression, behavioral symptoms, changes in the brain should be mentioned.

The Discussion section mentions 13 models with mutations in genes (line 659), but only two are given in this section. Please consider other models.

Section 1.3.3. Please add more information about experimental animals. Is this model used only to study the effects of pesticides or has broader applications?

Section 1.5. should be included in section 2

Line 233. The relationship between infection and the development of PD is not obvious. Viruses may be possible triggers for PD, but infection of mice should not be considered a model for studying PD.

Section 2. Please add a figure summarizing the interaction of the immune system and neuroinflammatory processes.

Author Response

Responses to Reviewer 1

I thank the authors for this interesting review. Unfortunately, it needs significant improvement. The authors should consider the relevance of each animal model to human’s Parkinson's disease. It is necessary to describe the correspondence between the processes occurring in humans and the processes in model animals.

The authors need to work on the consistency of presentation of the material. It may be worth highlighting two main parts: (1) Toxin-induced and genetic models (2) Methods for studying various disorders in PD (Neuroinflammation, immune dysfunction, Tissue infiltration and others).

The lack of illustrations is a significant drawback. Please add a figure that summarizes information about all PD models.

Thanks for your useful comments. On your advice, we have restructured the text of the manuscript. We have included in this revised manuscript Figure 1, which summarizes all experimental animal models related to immune dysfunction in Parkinson's disease.

Specific Comments:

Please expand the introduction section. Please indicate the main mechanisms of development of PD in humans, on the basis of which animal models were created. The purpose of this review should also be stated.

Section 1.1. More detailed information about α-synuclein, Lewy bodies, non-dopaminergic pathways and their role in PD should be provided. Please add a picture.

We have reorganized the manuscript text and expanded the introduction. In this revised version of the manuscript, the purpose of this review is at the end of the introduction.

Figure 1 is included in the revised manuscript summarizing all experimental models related to immune dysfunction in Parkinson's disease.

Section 1.3. Why are 6-OHDA and MPTP used to create animal models? What is the mechanism of action of these substances?

Thank you for your comment and suggestion. We have added the parts about the main mechanism of action of both toxins, 6-OHDA and MPTP. In the revised manuscript, they are now in the new sections 2.1.1. and 2.1.2.

Sections 1.3.1.1. and 1.3.1.2. Please describe the 6-OHDA-model and MPTP-model in more detail: symptoms of the disease, progression of the disease, MRI brain or brain slices, behavioral characteristics and more. The advantages and disadvantages should also be noted in these sections.

Thank you for your valuable comment. We have added the most important points about the course of the disease and the behavioral features that are within the scope of our manuscript. We have provided only the key points and details necessary to understand the scientific advantages and disadvantages of each model.

In section 1.3.1.3. the study of the role of immune cells in the development of PD is mentioned. What is known about their role in disease progression? Please add information about this in humans with PD and in animal models

We have included the information on the role of immune cells in MPTP in the new section 2.1.3, while the role of immune cells in 6-OHDA was not studied in extenso due to the nature of the induction.

Section 1.3.2. Please tell us in more detail about the mechanisms of PD occurrence with overexpressing α-syn and LRRK2 gene mutation. Please add information about experimental animals with these mutations: disease progression, behavioral symptoms, changes in the brain should be mentioned.

The Discussion section mentions 13 models with mutations in genes (line 659), but only two are given in this section. Please consider other models.

Thank you for your valuable comment.

The section 1.3.3 (now 2.2.) is an introduction of PD genetic models, which are discussed throughout the review. The description of these models as PD models is not the objective of this review. Rather, the focus is to introduce animal models that have been used to study the role of the immune system in PD, see the title of this section: “Animal models to study the immune system in PD”. It is through the following sections of the manuscript that details are given regarding specific results obtained with these models. To clarify this, we have added the following sentence at the beginning of Section 2: “The most commonly used animal models in the study of the immune system in PD are briefly described below."

In addition, following the reviewer's recommendation, we have added some more detail about the most commonly used genetic models: “Regarding genetic models, SNCA transgenic models do not show extra-nigral pathology, whereas LRRK mice show a slight increase in serotonin levels in the prefrontal cortex and a decrease in olfactory bulb dopaminergic neurons and locus coeruleus noradrenergic neurons at 24 months of age, and GBA mutant mice develop cholinergic dysregulation in the hippocampus (Lama et al., 2021)”

In this section, it is mentioned that there are models related to SNCA, LRRK2, GBA, but the number of mutations or models generated in relation to these genes is not specified. The discussion mentions that there are 13 different models related to SNCA. (These are not contradictory data).

Section 1.3.3. Please add more information about experimental animals. Is this model used only to study the effects of pesticides or has broader applications?

Thank you for your comment. Section 1.3.3 (now 2.3.), similar to the previous section, serves as an introduction and specifies the agrochemical toxic models (rotenone, PQ) used as models for inducing PD and their consequences. The objective of this review is not to extensively develop these models; rather, the aim of this section is to introduce them. Subsequently in the review, their use in studying the immune system in PD will be discussed.

Section 1.5. should be included in section 2

In this revised version of the manuscript, this section has been moved to the introduction, and the aim of this review is now at the end of the introduction.

Line 233. The relationship between infection and the development of PD is not obvious. Viruses may be possible triggers for PD, but infection of mice should not be considered a model for studying PD.

In this revised version of the manuscript, this point is explained in more detail – now in section 3.1 Adaptive immunity (Peripheral immunity in the pathogenesis of PD) the last two paragraphs.

Section 2. Please add a figure summarizing the interaction of the immune system and neuroinflammatory processes.

Figure 2 in this revised manuscript schematically summarizes the role of peripheral and central immunity in the pathogenesis of PD.

Reviewer 2 Report

Comments and Suggestions for Authors

The review discusses Parkinson's disease (PD) as a chronic, progressive condition associated with neuroinflammation and immune dysfunction. It examines methodological approaches to studying changes in central and peripheral immunity in PD, assessing their advantages, limitations, and applicability to humans. While no single animal model replicates all human PD features, neuroinflammation is commonly observed and crucial for understanding immune system involvement. The immune system's interactions within the central nervous system are pivotal in PD pathogenesis. Despite limitations, culture models aid in understanding immune interactions and disease progression. Identifying immune-mediated mechanisms could unveil therapeutic targets. The text underscores the importance of methodological guidelines for experiments with animal models and PD patients to ensure result validity and consistency.

Comments:

1. Authors should consider adding a table for all the possible cell culture models available to study immune dysfunction and neuroinflammation, along with their advantages and limitations.

2. The review acknowledges the role of genetic models in PD. However, it falls short of fully exploring how these models can be effectively utilized to investigate immune dysfunction in PD.

Author Response

Responses to Reviewer 2

The review discusses Parkinson's disease (PD) as a chronic, progressive condition associated with neuroinflammation and immune dysfunction. It examines methodological approaches to studying changes in central and peripheral immunity in PD, assessing their advantages, limitations, and applicability to humans. While no single animal model replicates all human PD features, neuroinflammation is commonly observed and crucial for understanding immune system involvement. The immune system's interactions within the central nervous system are pivotal in PD pathogenesis. Despite limitations, culture models aid in understanding immune interactions and disease progression. Identifying immune-mediated mechanisms could unveil therapeutic targets. The text underscores the importance of methodological guidelines for experiments with animal models and PD patients to ensure result validity and consistency.

Comments:

  1. Authors should consider adding a table for all the possible cell culture models available to study immune dysfunction and neuroinflammation, along with their advantages and limitations.

Thank you for your valuable comment. In accordance with the reviewer's recommendations, we have added a summarizing table showing the advantages and disadvantages of the different cellular models discussed in the review. Please refer to Table 1.

  1. The review acknowledges the role of genetic models in PD. However, it falls short of fully exploring how these models can be effectively utilized to investigate immune dysfunction in PD.

Thank you for your comment. However, the aim of this review is not to discuss specific experiments or models, but rather to mention the possible methodological options for studying the immune system in Parkinson's disease.

In particular, the section on cellular models mentions the utility of these genetic models for studying the immune system in Parkinson's disease:

  • Microglial primary culture of genetic models for synuclein or PINK1 are mentioned in 5.2.1 section.
  • Peripheral immune cells can be isolated from these genetic models to study their behavior or peripheral immune cells can be introduced to observe their neuroprotective or pro-inflammatory effects in these models.
  • Co-culture of microglia (immune system brain resident cells) and /or astrocytes with DA neurons from animals’ models is described as a model to study the interplay between cells related to inflammation and oxidative stress in the context of PD
  • Co-culture of peripheral immune cells from these PD models with brain-derived cells such as neurons or microglia helps to study the effects of immune cell infiltration on neuronal health and inflammation in the CNS.
  • Co-Culture of neurons derived from IPSCs and immune cells from animals models (or patients), is another example of the interaction of the immune system in Parkinson disease.
  • Brain slice cultures from animal models can be used to study the complexity of cellular interactions in the mesencephalon or striatum. In addition, these slices could be co-cultured with immune cells and provide valuable information on the interaction between the two factors.
  • Organoid derived from IPSCs of patients or animal models could be used to study the interaction of SI in PD, where these could be grown in coculture with IS cells.

As the title of the manuscript” "Experimental models to study immune dysfunction in the pathogenesis of Parkinson’s disease” makes clear, our approach in this review was to review the current state of the literature on the models and methods to study immune dysregulation in animal models of Parkinson's disease.

Round 2

Reviewer 1 Report

Comments and Suggestions for Authors

The new additions to the manuscript made a big difference. The quality of the paper had improved, and all my questions were addressed. No more comments.